# Prevalence and associated factors of diabetes among adult populations of Hawassa town, southern Ethiopia: A community based cross-sectional study

Getu Kassa Belete[1,2]*, Hlupheka Lawrence Sithole[2]

**1** World Health Organization (WHO), Hawassa, Sidama, Ethiopia, **2** College of Health Studies at the University of South Africa (UNISA), Limpopo, South Africa

* getish98@gmail.com

## Abstract

### Introduction

Knowing the magnitude and preventable risk factors of diabetes has a significant contribution in targeted prevention intervention which ultimately ensures the existence of healthier and productive individuals in a country. Diabetes has untoward impact on health, social and economic consequences. Exploring preventable risk factors are extremely important because of their potential association and interaction with diabetes. Therefore, the aim of this study was to investigate the magnitude and modifiable risk factors of diabetes among adult populations in Hawassa town, southern Ethiopia.

### Methods

A community based cross-sectional study was conducted from September, 2023 to November, 2023 among adult populations. A multi-stage sampling technique was employed to select 1,113 study participants between the ages of 20–69 years. An interviewer-administered questionnaire was used to collect data. Additionally, participants were also instructed to fast overnight, after which the standard fasting blood glucose test was conducted. A binary logistic regression model was fitted to identify independent predictors of diabetes.

### Results

The overall prevalence of diabetes was 14.4% (95% Confidence Interval (CI): 12.4%, 16.4%). Being male (Adjusted Odds Ratio (AOR):2.10; 95% CI: 1.34, 3.29), being unable to read and write (AOR: 3.38; 95% CI: 1.09, 10.47), read and write (AOR: 3.38; 95% CI: 1.09, 10.47) and medium cycle (AOR 2.79; 95% CI: 1.02, 7.63) compared to college and above, consume less than 5 servings of fruits on daily base (AOR: 2.80; 95% CI: 1.18, 6.62), having ever chewed khat (AOR 6.50; 95% CI: 4.07, 10.39) and being overweight and obese (AOR: 2.43; 95% CI: 1.54, 3.83) were independently associated with diabetes mellitus (DM).

**Data availability statement:** All data underlying the findings described in our paper are uploaded in Zenodo with https://zenodo.org/uploads/8284584.

**Funding:** The author(s) received no specific funding for this work.

**Competing interests:** The authors declare that they have no competing interests.

## Conclusion

This study identified a high prevalence of diabetes among adults in Hawassa, driven by various risk factors. This presents an opportunity to mitigate diabetes risk through public health measures, including avoiding khat chewing, promoting healthy diets, managing overweight and obesity, implementing community-based screening, enhancing health literacy, and integrating health information into daily life.

## Introduction

Globally, non-communicable disease like diabetes has increased and brought a tremendous economic burden to a nation and at individual level. It is projected that the number of individuals with diabetes will rise from 171 million in 2000 to 300 million by 2025 [1]. Moreover, this burden is projected to increase to 700 million by 2045 [2]. Most importantly, the prevalence of type 2 diabetes is estimated to rise to 11.3% by 2030 and 12.2% by 2040 globally [3]. Most painfully, 464 (9.1%) million and 298 million (5.8%) adults globally had impaired glucose tolerance and impaired fasting glucose level in 2021 respectively [4]. Besides, type 2 diabetes listed out in the seventh level among the top diseases that causes of disability and years of life lost (DALYs) globally [5]. Also, type 2 diabetes incidence increased from 8.5 million in 1990 to 21.7 million in 2019 [6]. Similarly, in 2017, type I diabetes was projected to account for around 2% of all instances of diabetes worldwide, with rates varying from less than 1% in Pacific countries to over 15% in Northern European populations [7]. Also, type I diabetes among US adults reported at 9.7% [8].

According to Alowayesh et al [9], the economic burden of diabetes was extremely huge in terms of drug expenditure. For instance, in 2018, the overall, annual expense for diabetic medicine in Kuwait was close to $201 million and at individual level amounted to $1,236.30. Worldwide, about 239.7 million (44.7%) individuals were unaware of their diabetes status and the biggest and the least proportion reported at 53.6% and 24.2% from Africa and North America and Caribbean, respectively [10]. As reported by US Department of Health and Human Services [11], 10% of United States people had diabetes. In this particular report, 7.3 million adults greater than 18 years were unaware of their diabetes status. Most painfully, in this particular study, the magnitude of diabetes in immigrants from Asia and Africa reported at 15% [12].

Likewise, overweight and obesity, tobacco use, alcohol consumption, khat chewing, dietary habits and physical inactivity were risk factors for diabetes mellitus. For instance, the magnitude of overweight and obesity were 36.3% and 18%.8% among type 2 diabetes patients in Sidama region, Ethiopia respectively [13]. Globally, over 200 million mortalities have been due to smoking cigarette in the past 3 years and 1.14 million persons were current smoker [14]. As reported by Fentaw, Fenta and Biresaw [15], the uppermost magnitude of alcohol use was found to be 76.7%, 68.7% and 65.6% in Mozambique, Ethiopia and Uganda in a pooled survey done in eleven east Africa countries respectively. Globally, 23% of the world's adult population is inadequately active and the proportion of physical inactivity mounted to 33% in middle-income countries [16]. Moreover, in a study conducted in Easter Mediterranean region revealed inadequate intake of fruits and vegetables which was less than 280g on daily bas [17]. Besides, in another survey done in Africa region showed the consumption of fruits and vegetables at 356g per day (below the recommended target 400g per day) [18]. Also, the magnitude of khat chewing in Hossana, Ethiopia reported at 58%, of those khat chewers, 75.2% and 24.7% were men and women, respectively [19]. In addition, the prevalence of khat chewing with type 2 diabetes mellitus in Saudi Arabia was 29.3% [20]. Also, a different study

conducted in Yemen showed association of khat chewing with type 2 diabetes mellitus and pre-diabetes [21].

Sub-Saharan Africa is estimated to have a sharpest rise in the magnitude of diabetes in the coming 25 years. For instance, in Mozambique, recently the magnitude of diabetes reported at 7.4% which was higher than the finding in 2005 (2.9%) [22]. In addition, the prevalence of diabetes in study participants aged 20 to 69 in Botswana was reported to be 9.3% [23]. In Zanzibar, the prevalence of diabetes was a startling 4.4%, and it was likewise associated with older adults and higher body weight [24]. Similarly, diabetes was reported to affect 36.5% of Somalian high school and university instructors, as reported by Ibrahim et al.[25]. Most importantly, the magnitude of diabetes mellitus reported in Ethiopia at 3.3% and the proportion of diabetes mellitus was 2.3% and 4.6% for individuals living in countryside and town, respectively [26]. As reported by Assefa et al [27], the prevalence of diabetes mellitus among the ostracized Menja community in South West Ethiopia reported at 14.7% (95% CI: 11.1–18.3), the highest prevalence observed in men than women, 16.8% versus 11.1% correspondingly. In a different survey done in a similar country, the magnitude of diabetes reported at 7% [28]. Likewise, close to 4.7% of Ethiopian population were reported to have diabetes in the year 2021 [29]. Also, 12.2% of patients were reported to have diabetes in Hawassa Comprehensive Specialized hospital [30]. A previous study conducted in Hawassa town was facility-based and mainly focused on individuals seeking medical care. In contrast, this study was aimed to determine the prevalence of diabetes and risk factors among adults through a community-based survey in Hawassa town, southern Ethiopia.

## Materials and methods

### Study setting

The study was conducted in Hawassa town, Sidama regional state, Southern Ethiopia. Hawassa is located 273 km south of Addis Ababa, the capital city of Ethiopia. The town has a total population of 399,461. There are 32 *kebeles, or* the smallest administrative units in Ethiopia, and eight sub-cities in town. There are also 81,523 households in the town.

### Study design and period

A community based cross-sectional study design was employed from September, 2023 to November, 2023 in Hawassa town, southern Ethiopia.

### Population

The source population for this particular study was all adult populations living in Hawassa town. Also, the study population was all adult population between age 20- and 69-years old residing in eight *kebeles* in Hawassa town. The study subjects were individuals between age 20 and 69 years selected randomly at household level.

### Inclusion and exclusion criteria

All adults between age 20 and 69 years inhabiting in the Hawassa town for minimum 6 months preceding this survey was included in the study. Furthermore, previously diagnosed and known diabetic patients were included in the study. Pregnant women, physical disabilities, terminally sick and individual with mental illness excluded from this survey.

### Sample size determination and sampling technique

A single population proportion formula was employed to estimate the study sample size to determine the magnitude of diabetes and risk factors among adult population of Hawassa

town. The subsequent assumptions were considered while calculating the sample size: A prevalence of diabetes 12.4% [31], a 5% margin of error, a design effect of 1.5, and a 10% non-response rate. Likewise, sample size was calculated for other associated factors of diabetes for comparison as shown on Table 1. Therefore, the final sample size for this specific survey was 1,113. Multi-stage sampling with age stratification method employed in this survey. A four-stage survey was conducted. First, Sample was allocated proportionally to the eight sub-cities based on population number. Secondly, using random sampling, one *kebele* selected from each sub-cities. The *kebeles* included in this study were Daka, Gudumale, Addis Abeba, Harer, Teso, Guwi, Tilte and Chefe. Thirdly, using systematic random sampling a required number of households from each *kebeles* were selected. Likewise, the $K^{th}$ household identified by dividing the whole population of the village (N) to the allocated sample size (n) for that specific village ($k^{th} = N/n$) and the subsequent households were selected using systematic random sampling from the household sampling frame (Fig 1). Finally, a study subject selected randomly using the kish method (random selection of study subjects from eligible age groups at household level) among the eligible groups between 20–69 years old [32]. During the first visit to the household, an interview was done. The second visit was made after instruction for overnight fasting or eight hour fasting prior to blood collection.

## Study variables

Prevalence diabetes was the dependent variable, while the independent variables were body mass index, diet, khat usage, alcohol use, cigarette use, ethnicity, religion, age, sex, marital status, educational attainment, and employment status.

## Definition of terms

**Body Mass Index (BMI):** A standard cut-offs limit calculated using the formula $kg/m^2$ to assess an individual's body mass and classified as underweight ($<18.5\,kg/m^2$), normal weight ($18.5–24.9\,kg/m^2$), overweight ($25–29.9\,kg/m^2$), and obesity ($\geq 30\,kg/m2$).

**Currently tobacco smoking:** Self-reported currently tobacco smoking.

**Diabetes:** It is defined as a fasting blood glucose level of $\geq$ 126 mg/dL (or >7 mmol/L), a self-reported diagnosis of diabetes, or the use of oral or injectable hypoglycemic agents.

**Ever chewed khat:** Self-reported ever khat chewed.

**Ever consumed alcohol:** Self-reported ever consumed alcohol.

**Greater than 5 servings of fruits and vegetables:** Consuming greater or equal to 5 platefuls (400 gm) on daily base.

**Table 1. Sample size determination using open Epi software program.**

| Variables | Percentage of control exposed to the risk factor | Adjusted Odds Ratio (AOR) | Sample size | Sample size after 1.5 design effect | Final sample size after adding 10% for non-response rate | Reference |
|---|---|---|---|---|---|---|
| Body Mass Index (Normal) | 6.32 | 9.2 | 64 | 96 | 106 | Seifu, Y et. al, 2020 [31] |
| Smoking (No) | 6.73 | 7.8 | 74 | 111 | 122 | Seifu, Yet. al, 2020 [31] |
| Exercise (Yes) | 8.76 | 3.0 | 250 | 375 | 413 | Kassa, A et.al, 2019 [30] |
| Fruits and vegetable servings/day (Adequate) | 6.9 | 2.8 | 350 | 525 | 578 | Feyisa, B et.al, 2022 [28] |
| Hypertension (Normal) | 6.3 | 2.9 | 352 | 528 | 581 | Feyisa, B et.al, 2022 [28] |

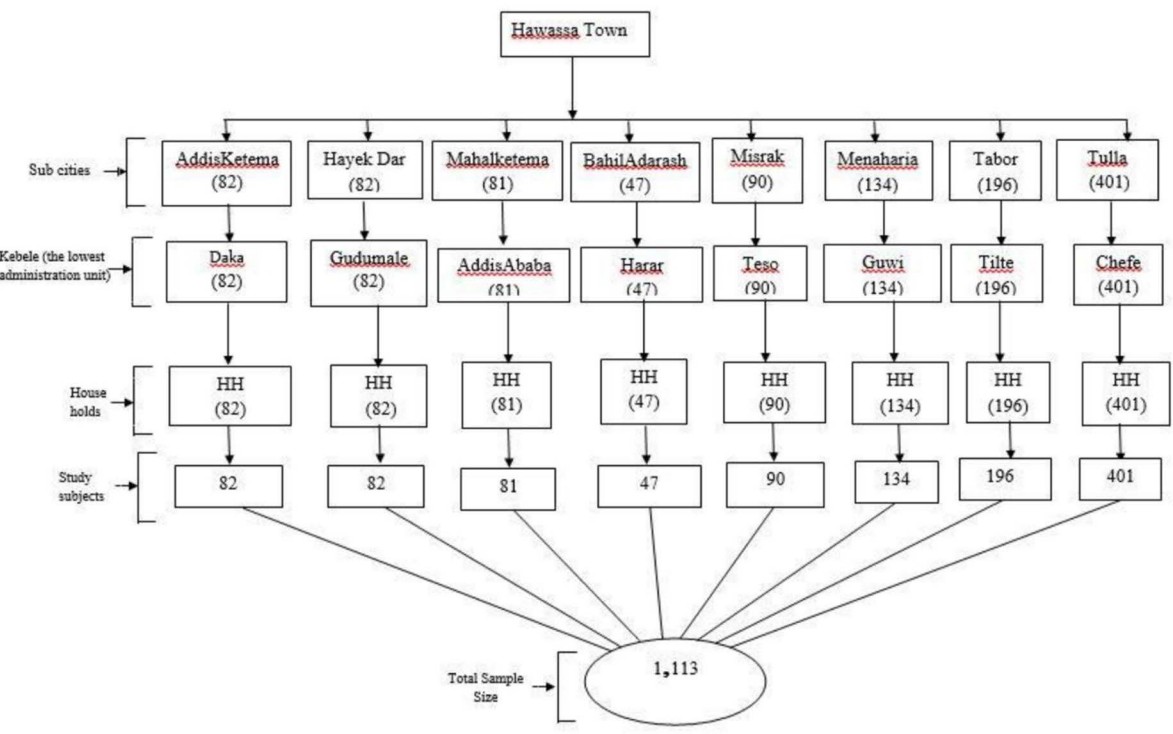

**Fig 1. Sampling technique for selection of study subjects among adult population of Hawassa town, Ethiopia.**

**Insufficient exercise:** Inadequate physical exercise less than 30 minutes in a day or less than 150 minutes in a week.

**Kish methods:** Random selection of study subject from a household.

**Less than 5 servings of fruits and vegetables:** Consuming below 5 platefuls (400gm) on daily base.

**Moderate physical exercise (brisk walking):** Moderate increase in breathing (brisk walking or carrying light loads) for 30 minutes per day continuously.

**Non-communicable diseases:** Are non-infectious diseases with prolonged course.

**Normal weight:** Body mass index measured between 18.5 kg/m² and 24.9 kg/m²

**Obese:** Body Mass Index measured greater than or equal to 30 kg/m².

**Overweight:** Body mass index measured between 25 kg/m² and 29.9 kg/m².

**STEPS approach:** Is a successive step that begins with collecting major information on risk factors using a questionnaire, followed by blood collection for blood glucose test.

## Data collection

Data collection instrument was adapted from the World Health Organization non-communicable diseases surveillance guideline (WHO STEPs surveillance guideline). A standard glucometer (Auto-code) was used for fasting blood glucose test following overnight fasting prior to testing. The glucometer calibration was done with instrument code key both

before the blood test and on a frequent basis to ensure the accuracy of the glucometer during the data gathering time.

The researcher introduced data collection tool to data collectors and supervisors. Furthermore, data collectors and supervisors were trained for 2 days before deployment for data collection. In the training, data collectors obtained skills how to conduct interview and fasting blood glucose test. All data collectors were nurses. The data collection instrument was primarily in English, then translated to the local Amharic language for easier understanding using professional translators, then translated again to English. Likewise, a robust monitoring employed to ensure data completeness. Moreover, the data collection tool was validated in pilot study in adjacent villages which were not included in the study. Experts with extensive experience evaluated the tool's validity, and questions that were deemed unclear were adjusted accordingly. Additionally, on a pre-test, 5% of the sample size was examined. Because the questioner measured what it was supposed to measure, it was valid. We tested before the study participants eat their breakfast. Additionally, data collectors strongly recommended fasting for the entire night. Additionally, it was often questioned if individuals had fasted before the exam. We recommended that research participants take a HgbA1c test two to three months later. However, due to resource constraints, we do not conduct testing after two to three months.

Measurements BMI were taken using a portable digital weight scale and height measuring instrument (Seca). Daily calibration was carried out with a standard kilogram weight. In addition, the instrument was positioned nearer a wall on a level floor. The device was tested for satisfactory operation, pointing at zero reading, before any data was collected. In addition, study participants were instructed to remove overcoats and jackets, stand erect on a beam facing the data collectors, and go barefoot. The correct protocol for disposing of waste was adhered to when gathering the data. Waste materials were collected in plastic bags and disposed of at the neighboring health facilities' incinerators for burning.

## Fasting blood glucose test

Digital glucometer (Prodigy AutoCode OK Biotech Co. Ltd, Hsinchu city, Taiwan) was used to measure fasting blood glucose. A whole blood was used to measure fasting blood glucose level. Aseptic technique was employed to get a tiny blood from a ring finger. A slight finger prick was done using sterile lancet. A tiny of blood allowed touching a test strip after inserting in a digital auto-code. Thereafter, a data collector sees and hears accurate result after 7 seconds. The data collectors recorded all the findings of fasting blood glucose measures on the questionnaires. The newly diagnosed diabetic patients were advised on lifestyle modification. Moreover, they were told to make medical follow up at the nearby health facility at regular base.

## Statistical analysis

Data entered in SPSS software with version 26.0. Data was submitted to supervisors and checked for completeness and accuracy on daily base. Furthermore, data was checked and cleaned prior to analysis to maintain data quality. Likewise, SPSS software with version 26 was utilized to analyze frequency, compare dependent and independent variables. In this survey, age, sex, marital status, educational status, ethnicity, employment status, religion, tobacco smoking, alcohol use, physical inactivity, diet, Khat chewing, and Body mass index are independent variables. On the other hand, diabetes is dependent variable. Socio-demographic data was analyzed in the first phase, behavioural and biological risk factors analysis was done secondly. Furthermore, the result compared against the cut-offs of p-value less than 0.05.

Moreover, bi-variable and multi-variable logistic regression was done to figure out the independent predictor of the outcome. Variables with p-value < 0.25 in bi-variable analysis were inserted in multi-variable analysis to control confounding factors. Besides, OR (odds ratio) used for comparison at 95% C.I. Furthermore, the Hosmer-Lemeshow goodness-of-fitness model was checked and found to be 0.772. WHO AnthroPlus cut-offs used for comparison of the prevalence results of diabetes and risk factors. The findings of quantitative study displayed on tabular form to make comparison of one value against the other.

### Ethical consideration

The university of South Africa College of Human Science Research Ethics Committee approved and granted final ethics clearance to undertake the study (CREC Reference # 13652133_CREC_CHS_2023 Dated 31 July 2023). Written informed consent was obtained from all study participants.

## Results

### Socio-demographic characteristics

The overall number of study subjects was 1,113 with a response rate of 100%. The median age of the study subjects was 29 with interquartile range of 26–38 years. Of all the study subjects, 602 (54.1%) and 261 (23.5%) were between the age groups of 20–29 and 30–39 years, respectively. Likewise, among the study subjects 455 (40.9%) and 658 (59.1%) were men and women, respectively. In view of educational status, 291 (26.1%) and 237 (21.3%) were in medium cycle and high school educational level. With regard to ethnic groups, Amhara accounted for 474 (42.6%) followed by Sidama, 273 (24.5%). Also, of the study subjects, 672 (60.4%) and 380 (34.1%) were never married and currently married, respectively. With regard to religion, 727 (65.3%) were protestant religion followers, followed by 278 (25.0%) orthodox religion believers. Likewise, 422 (37.9%), 263 (23.6%) and 217 (19.5%) study subjects were student, house wife and self-employed, respectively (see Table 2).

### Behavioural and biological risk factors of diabetes

The prevalence of tobacco smoking was 8.1% among the study subjects. Of the study subjects, 90 (8.1%) and 94 (8.4%) were currently smoking and ever smoked cigarette, respectively. Likewise, 39 (41.5%) and 38 (40.4%) smoked less than 5 and 5–10 cigarettes on daily base, correspondingly. Moreover, 48 (51.1%) and 30 (31.9%) of study subjects smoked cigarette less than 5 and 5–10 years, respectively. This survey also assessed alcohol consumption among study subjects of adult population. Accordingly, the magnitude of ever alcohol drinking of all types like beer, local drinks 'Araque', 'Tella' and 'Tej' was 22.1%. Likewise, 211 (85.8%) of the study subjects consumed alcohol in the past 12 months. Besides, regarding the frequency of alcohol consumption in the 12 months prior to the survey, 52 (21.1%) consumed daily; 35 (14.2%) consumed 5–6 days per week; 23 (9.3%) consumed 3–4 days per week and 42 (17.1%) consumed 1–2 days per week. Among the study subjects 259 (23.3%) involved in vigorous-intensity activity that causes huge increase in breathing at least for 10 minutes. Besides, moderate-intensity activity that causes small increase in breathing reported at 277 (24.9%). Notably, nearly 43.4% were able to practice brisk walking greater than 30 minutes in a day. Most painfully, about 1012 (90.9%) and 1010 (90.7%) of study participants had less than 5 servings of fruits and vegetables respectively (WHO recommend ≥ 5 servings of fruits and vegetables on daily base). Likewise, study subjects ever chewed khat accounted for 217 (19.5%) (see Table 3).

**Table 2. Socio-demographic characteristics of the study subjects of adult population in Hawassa town, Ethiopia (n = 1,113).**

| Variables | Number | Percent |
|---|---|---|
| **Sex** | | |
| Male | 455 | 40.9 |
| Female | 658 | 59.1 |
| **Age** | | |
| 20–29 | 602 | 54.1 |
| 30–39 | 261 | 23.5 |
| 40–49 | 139 | 12.5 |
| 50–59 | 76 | 6.8 |
| 60–69 | 35 | 3.1 |
| **Educational status** | | |
| Unable to read and write | 173 | 15.5 |
| Read and write | 99 | 8.9 |
| First cycle completed (1–6 grade) | 181 | 16.3 |
| Medium cycle completed (7–8) | 291 | 26.1 |
| High school (9–12) | 237 | 21.3 |
| College and above | 132 | 11.9 |
| **Ethnicity** | | |
| Oromo | 105 | 9.4 |
| Amhara | 474 | 42.6 |
| Sidama | 273 | 24.5 |
| Gurage | 99 | 8.9 |
| Wolaita | 102 | 9.2 |
| Others | 60 | 5.4 |
| **Marital status** | | |
| Never married | 672 | 60.4 |
| Currently married | 380 | 34.1 |
| Separated, divorced and widowed | 61 | 5.5 |
| **Religion** | | |
| Orthodox | 278 | 25.0 |
| Muslim | 68 | 6.1 |
| Protestant | 727 | 65.3 |
| Others | 40 | 3.6 |
| **Employment status** | | |
| Government employee | 105 | 9.4 |
| Self-employed | 217 | 19.5 |
| Student | 422 | 37.9 |
| House wife | 263 | 23.6 |
| Farmer | 54 | 4.9 |
| Retired and unemployed | 52 | 4.7 |

## Prevalence of diabetes

The prevalence of diabetes was 14.4% (95% CI: 12.4%, 16.4%) (Fig 2). Likewise, the prevalence of diabetes was 20.7% in male and 10.0% in female study subjects (Fig 3). Also, the median age for diabetic was 29.0 years with interquartile range (IQR) of 26–38 years. Besides, the median value of fasting blood glucose measure was 97.0 gm/dl with interquartile range of 89.0–110.0

**Table 3. Magnitude of behavioural and biological risk factors of diabetes among adult population of Hawassa town, Ethiopia.**

| Variables | Number | Percent |
|---|---|---|
| **Current smoking cigarette or pipe** | | |
| Yes | 90 | 8.1 |
| No | 1023 | 91.9 |
| **Currently smoke tobacco daily** | | |
| Yes | 90 | 8.1 |
| No | 1023 | 91.9 |
| **Ever smoked cigarettes** | | |
| Yes | 94 | 8.4 |
| No | 1019 | 91.6 |
| **Number of cigarettes smoke daily** | | |
| Less than 5 | 39 | 41.5 |
| 5–10 | 38 | 40.4 |
| Greater than 10 | 17 | 18.1 |
| **Number of years smoke cigarette** | | |
| Less than 5 | 48 | 51.1 |
| 5–10 | 30 | 31.9 |
| Greater than 10 | 16 | 17.0 |
| **Ever consumed any alcohol** | | |
| Yes | 246 | 22.1 |
| No | 867 | 77.9 |
| **Consumed alcohol in the last 12 months** | | |
| Yes | 211 | 85.8 |
| No | 35 | 14.2 |
| **Consumed at least one standard alcohol in the past 12 months** | | |
| Daily | 52 | 21.1 |
| 5–6 days per week | 35 | 14.2 |
| 3–4 days per week | 23 | 9.3 |
| 1–2 days per week | 42 | 17.1 |
| 1–3 days per month | 31 | 12.6 |
| Less than once a month | 28 | 11.4 |
| Never | 35 | 14.2 |
| **Vigorous-intensity activity that causes large increase in breathing** | | |
| Yes | 259 | 23.3 |
| No | 854 | 76.7 |
| **Moderate-intensity activity that causes small increase in breathing** | | |
| Yes | 277 | 24.9 |
| No | 836 | 75.1 |
| **Moderate exercise or (brisk walking) in a day** | | |
| < 30 minutes | 142 | 65.6 |
| > or = 30 minutes | 109 | 43.4 |
| **Servings of fruits per day** | | |
| Less than 5 | 1012 | 90.9 |
| Greater or equal to 5 | 101 | 9.1 |
| **Servings of vegetables per day** | | |
| Less than 5 | 1010 | 90.7 |
| Greater or equal to 5 | 103 | 9.3 |

*(Continued)*

**Table 3.** (Continued)

| Variables | Number | Percent |
|---|---|---|
| **Ever chewed khat** | | |
| Yes | 217 | 19.5 |
| No | 896 | 80.5 |
| **Body Massa Index** | | |
| Normal | 861 | 77.4 |
| Overweight and obese | 252 | 22.6 |
| **Hypertension** | | |
| Yes | 181 | 16.3 |
| No | 932 | 83.7 |

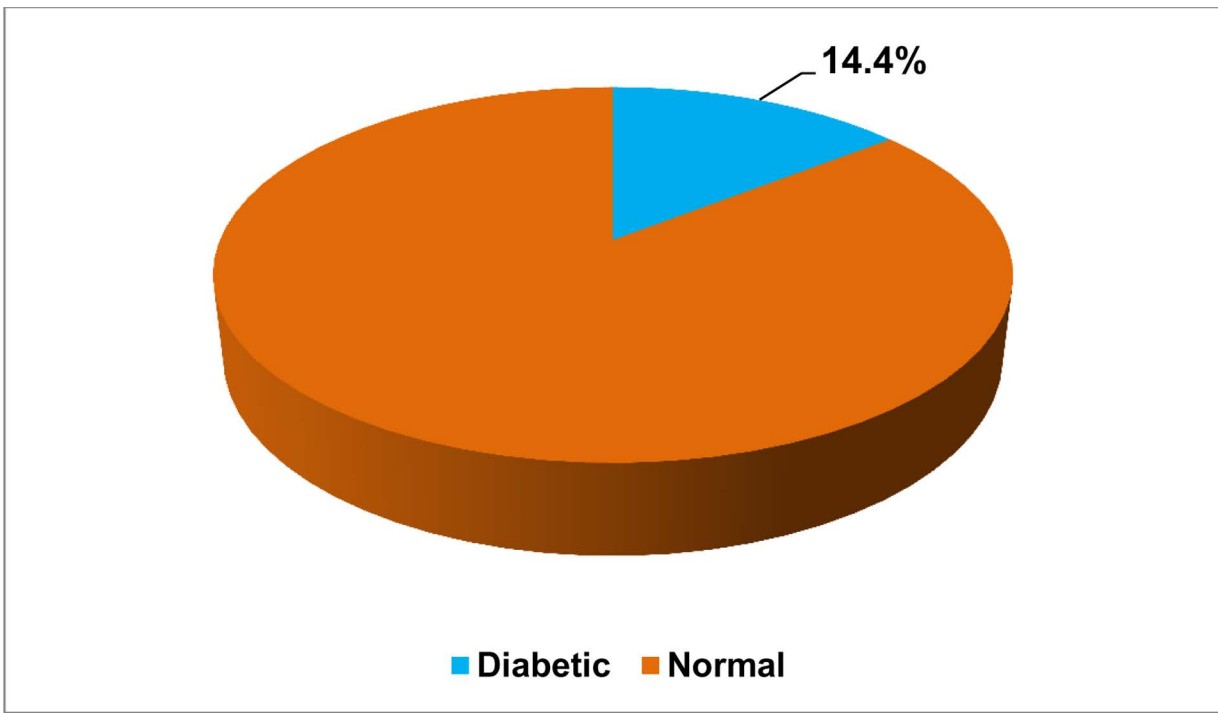

**Fig 2. Prevalence of diabetic among the study subjects of adult population in Hawassa Town, Ethiopia, 2023.**

gm/dl. The number of newly diagnosed DM patients were 101 (63.1%). The number of known DM cases were 59 (36.9%). The prevalence of pre-diabetes was 283 (26.9%) among study participants excluding known diabetes patients from the denominator.

## Factors associated with diabetes mellitus

Variables such as sex, age, education, employment, current smoking, ever consumed alcohol, ever chewed khat and BMI were significantly associated with diabetes in bi-variable binary logistic regression. However, only sex, education, ever chewed khat, BMI and hypertension were independent predictors of DM in multi-variable analysis after controlling other confounding variables.

The odds of having diabetes were 2.10 times higher in males than their counterpart, with (95% CI: 1.34, 3.29). Most importantly, being unable to read and write (AOR: 4.38; 95% CI:

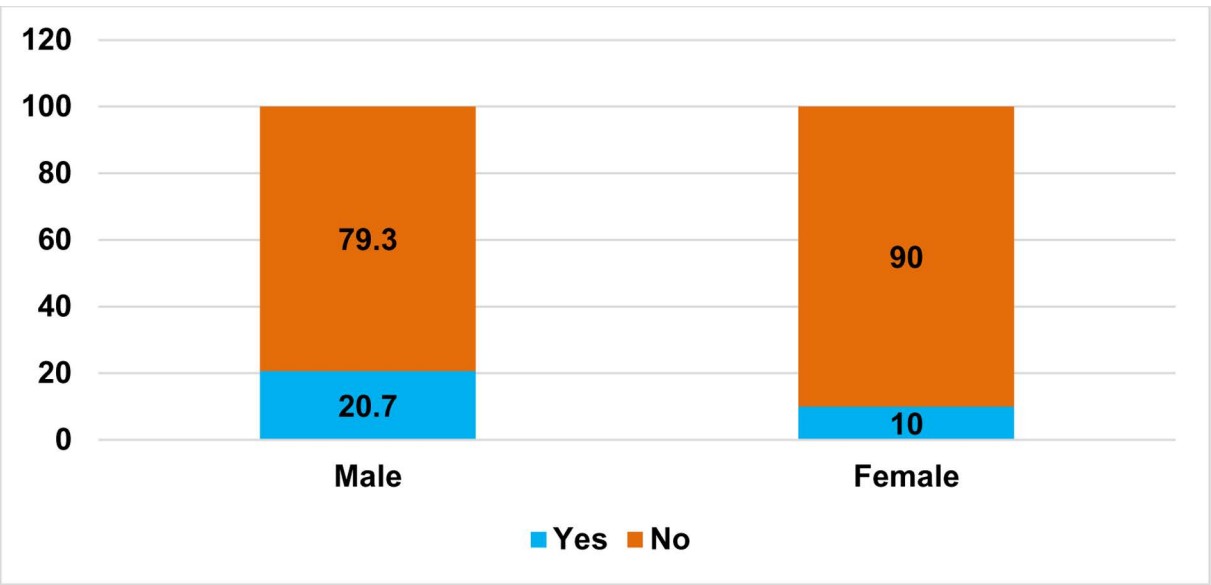

**Fig 3. Proportion of diabetes in males and females among adult population in Hawassa Town, Ethiopia, 2023.**

1.49, 12.86), being read and write (AOR: 3. 58; 95% CI: 1.14, 11.21), being medium cycle (AOR: 2.87; 95% CI: 1.04, 7.91) educational status and hypertension (AOR: 2.55; 95% CI: 1.62, 4.01) were independently associated with diabetes. Most importantly, the prevalence of diabetes was 15.0% among study participants who used to consume less than 5 servings of fruits compared to those consumed greater or equal to 5 servings (8.9%) on daily base. Most painfully, the prevalence of diabetes was 41.5% among study participants who ever chewed khat. The risk of having diabetes was (AOR: 6.31; 95% CI: 3.95, 10.08) times higher among ever chewed khat than those did not chew khat. Also, 19.0% of study participants with over-weight and obese value of body mass index found to be diabetic. Moreover, the risk of having diabetes among overweight and obese study participants was higher than normal body weight, with (AOR: 2.28; 95% CI: 1.44, 3.60). Likewise, 28.2% of study participants with diabetes had hypertension. The risk of having hypertension was (AOR: 2.55; 95% CI: 1.62, 4.01) times higher among diabetic cases than those did not have diabetes (see Table 4).

## Discussion

The prevalence of diabetes was reported at 14.4% (95% CI: 12.4, 16.4) among 20–69 years old adult population of Hawassa town. The finding is consistent with previous community-based studies conducted in southern Ethiopia, such as the 14.7% prevalence in the ostracized Menja community [27] and 12.4% in the Hawassa Zuria Woreda [31]. In contrast, the finding is higher than those of previous studies conducted in other countries, such as Botswana (9.3%), Zanzibar (4.4%) and Thailand (9.9%) [23],[24],[33]. The finding shows that diabetes may have become a public health burden among the adult population of Hawassa town. Also, higher prevalence of diabetes reported in the age category between 50–59 (19.7%) years and 30–39 (18.4%) years. Most importantly, this finding points out a prompt enrollment of cases for care and treatment, and ultimately strengthens screening for diabetes at community level. In this particular research, the prevalence of diabetes was higher in Males than females, 20.7% versus 10.0% respectively. In contrast, the finding is higher than those of previous studies conducted in Korea (males, 10.7% and females, 8.4%) [34] and Ethiopia (males, 9.1% and females, 8.4%)

**Table 4. Bi-variable and multi-variable analysis of factors associated with diabetes among adult population in Hawassa town, Ethiopia.**

| Variables | Diabetes | | COR (95%CI) | AOR (95%CI) |
|---|---|---|---|---|
| | Yes n (%) | No n (%) | | |
| **Sex** | | | | |
| Male | 94 (20.7) | 361 (79.3) | 2.34 (1.66,3.28)** | 2.10 (1.34,3.29)* |
| Female | 66 (10.0) | 592 (90.0) | 1 | 1 |
| **Age** | | | | |
| 20–29 | 76 (12.6) | 526 (87.4) | 1 | 1 |
| 30–39 | 48 (18.4) | 213 (81.6) | 1.56 (1.05,2.32)* | 1.44 (0.85,2.42) |
| 40–49 | 17 (12.2) | 122 (87.8) | 0.96 (0.55,1.69) | 0.96 (0.47,1.97) |
| 50–59 | 15 (19.7) | 61 (80.3) | 1.70 (0.92,3.15) | 1.43 (0.64,3.18) |
| 60–69 | 4 (11.4) | 31 (88.6) | 0.89 (0.31,2.60) | 0.47 (0.12,1.78) |
| **Educational status** | | | | |
| Unable to read and write | 36 (20.8) | 137 (79.2) | 5.52 (2.25,13.54)** | 4.38 (1.49,12.86)* |
| Read and write | 16 (16.2) | 83 (83.8) | 4.05 (1.52,10.77)* | 3.58 (1.14,11.21)* |
| First cycle (1–6) | 27 (14.9) | 154 (85.1) | 3.68 (1.47,9.20)* | 2.62 (0.91,7.53) |
| Medium cycle (7–8) | 44 (15.1) | 247 (84.9) | 3.74 (1.55,9.02)* | 2.87 (1.04,7.91)* |
| High school (9–12) | 31 (13.1) | 206 (86.9) | 3.16 (1.28,7.79)* | 2.59 (0.95,7.07) |
| College and above | 6 (4.5) | 126 (95.5) | 1 | 1 |
| **Ethnicity** | | | | |
| Oromo | 16 (15.2) | 89 (84.8) | 1 | |
| Amhara | 84 (17.7) | 390 (82.3) | 1.20 (0.67,2.14) | 1.06 (0.53,2.12) |
| Sidama | 30 (11.0) | 243 (89.0) | 0.69 (0.36,1.32) | 0.82 (0.37,1.79) |
| Gurage | 7 (7.1) | 92 (92.9) | 0.42 (0.17,1.08) | 0.61 (0.22,1.71) |
| Wolaita | 15 (14.7) | 87 (85.3) | 0.96 (0.45,2.06) | 0.81 (0.33,1.99) |
| Others¥ | 8 (13.3) | 52 (86.7) | 0.86 (0.34,2.14) | 0.95 (0.33,2.77) |
| **Employment status** | | | | |
| Government employee | 9 (8.6) | 96 (91.4) | 1 | 1 |
| Self-employed | 38 (17.5) | 179 (82.5) | 2.26 (1.05,4.88)* | 1.32 (0.52,3.39) |
| Student | 64 (15.2) | 358 (84.8) | 1.91 (0.92,1.97) | 1.55 (0.61,3.93) |
| House wife | 35 (13.3) | 228 (86.7) | 1.64 (0.76,3.54) | 1.84 (0.68,4.98) |
| Farmer | 9 (16.7) | 45 (83.3) | 2.13 (0.79,5.74) | 1.16 (0.32,4. 19) |
| Retired and unemployed | 5 (9.6) | 47 (90.4) | 1.14 (0.36,3.58) | 0.59 (0.14,2.52) |
| **Current smoking cigarettes or pipe** | | | | |
| Yes | 33 (36.7) | 57 (63.3) | 4.09 (2.56,6.52)** | 1.55 (0.87, 2.76) |
| No | 127 (12.4) | 896 (87.6) | 1 | |
| **Ever consumed any alcohol** | | | | |
| Yes | 72 (29.3) | 174 (70.7) | 3.66 (2.57,5.21)** | 1.36 (0.86,2.15) |
| No | 88 (10.1) | 779 (89.9) | 1 | |
| **Servings of fruits** | | | | |
| Less than 5 servings | 152 (15.0) | 860 (85.0) | 2.06 (0.99,4.32) | 3.24 (1.35,7.78) |
| Greater than 5 servings | 8 (7.9) | 93 (92.1) | 1 | |
| **Servings of Vegetables** | | | | |
| Less than 5 servings | 151 (15.0) | 859 (85.0) | 1.84 (0.91,3.72) | 1.13 (0.50,2.57) |
| Greater than 5 servings | 9 (8.7) | 94 (91.3) | 1 | |
| **Ever chewed Khat** | | | | |
| Yes | 90 (41.5) | 127 (58.5) | 8.36 (5.81,12.03)** | 6.31 (3.95,10.08)** |
| No | 70 (7.8) | 826 (92.2) | 1 | |

*(Continued)*

**Table 4.** (Continued)

| Variables | Diabetes | | COR (95%CI) | AOR (95%CI) |
|---|---|---|---|---|
| | Yes n (%) | No n (%) | | |
| **Body mass Index** | | | | |
| Normal | 112 (13.0) | 749 (87.0) | 1 | 1 |
| Overweight and obese | 48 (19.0) | 204 (81.0) | 1.57 (1.09,2.28)* | 2.28 (1.44,3.60)** |
| **Hypertension** | | | | |
| Yes | 51 (28.2 | 130 (71.8) | 2.96 (2.03, 4.33)** | 2.55 (1.62,4.01)** |
| No | 109 (11.7) | 823 (88.3) | 1 | |

¥ Others = Kembata and Silte ethnic groups;

* P value < 0.05;

** P value < 0.001COR = Crude Odds Ratio; AOR = Adjusted Odds Ratio, CI = Confidence Interval

[35]. The reasons could be due to individual level dietary habit, exercise, and family history. Likewise, the odds of having diabetes was 2.10 times higher in males than females (95% CI:1.34,3.29). This might be due to individual level lifestyle nature and family history of diabetes. In addition, the odds of having diabetes is higher among unable to read and write, read and write and medium cycle compared to college and above. This could be because individuals with low education status might have a limited knowledge regarding a healthier dietary habit and exercise which in turn result in obesity and eventually predispose to diabetes. This finding trigger to mitigate diabetes risk through public health measures, including promoting healthy diets, and managing overweight and obesity, towards a healthier life.

The prevalence of ever chewed khat is 41.5% among diabetic cases. In contrast, the finding is higher than those studies conducted in Ethiopia (19%) by Teklie et al. [36] and in Saudi Arabia (29.3%) by Badedi et al. [37]. The difference could be due to study participants' individual behaviour. Most importantly, in this particular study, prevalence of diabetes among overweight and obese study subjects reported at 19.0%. Conversely, the finding is lower than a study done in UK (25.7%) [38]. The discrepancy could be due to individual dietary habit and physical inactivity. Furthermore, overweight and obesity might attribute a significant role interfering with proper utilization of insulin in the body that eventually leads to diabetes. Most importantly, 4.7% of the study subjects had both diabetic and hypertension which is comparable with those studies conducted in China (3.8%) [39] and Vietnam (4.7%) [40]. In this case, the association between diabetes and hypertension might be due to the hemodynamic alteration that triggers renin-angiotensin-aldosterone system and changes in blood calcium level. Hence, diabetic has to be closely monitored to diminish morbidity and mortality among adult population of Hawassa town.

This study had a number of limitations. Interviewees might have eaten breakfast prior to fasting blood test. However, before testing, data collectors have frequently questioned respondents about whether or not they had eaten breakfast in an effort to close this gap. Furthermore, pregnant women could affect body mass index values. To minimize such bias, data collectors were able to question respondents about the status of pregnancy, menstrual cycle and amenorrhea prior to taking weight.

## Conclusion

In conclusion, this study identified a noticeable prevalence of diabetes and its association with modifiable risk factors among adult population of Hawassa town. The evidence-based

risk factors associated with diabetes highlight the need for urgent public health interventions, particularly community level screening programs aimed at early detection and prevention. In addition, the findings emphasize the importance of addressing behavioural practice, such as khat chewing, promoting lifestyle modifications, including dietary changes and the reduction of overweight and obesity. Regular check-ups and follow-up care for individuals with diabetes are also crucial. Moreover, risk communication strategies for diabetes should be integrated into public health campaigns to raise awareness. Lastly, these findings could serve as a baseline for future studies focused on operational study related to diabetes management and prevention.

## Acknowledgments

We would like to thank the University of South Africa (UNISA) for the permission to conduct the study. We also appreciate and thank Sidama regional health bureau, Sidama regional public health institution and Hawassa town health department for permitting the research to undertake. We are also very grateful to the Supervisor and coordinator in particular (Mr. Tekeba Habtewold and Dr. Tarekegn Solomon) for their day-to-day monitoring. We would also extend our appreciation to data collectors for their unreserved determination and job during data collection period. Lastly, but not least we acknowledge the study participants for their time and cooperation during data collection process.

## Author contributions

**Conceptualization:** Getu Kassa Belete.

**Data curation:** Getu Kassa Belete.

**Formal analysis:** Getu Kassa Belete.

**Methodology:** Getu Kassa Belete.

**Resources:** Getu Kassa Belete.

**Supervision:** Hlupheka Lawrence Sithole.

**Writing – original draft:** Getu Kassa Belete.

**Writing – review & editing:** Hlupheka Lawrence Sithole.

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
