## [Decision Letter · Decision Letter 0]

16 Jun 2024

PONE-D-24-07197Prevalence and risk factors of diabetes among adult populations of Hawassa town, Ethiopia: A community based cross-sectional study  Dear Dr. Belete,

Thank you for submitting your manuscript to PLOS ONE. After careful consideration, we feel that it has merit but does not fully meet PLOS ONE’s publication criteria as it currently stands. Therefore, we invite you to submit a revised version of the manuscript that addresses the points raised during the review process. could you please correct and amend the given comments as possible?

We look forward to receiving your revised manuscript.

Kind regards,

Mengistu Hailemariam Zenebe, PhD

Academic Editor

PLOS ONE

Journal Requirements:

The American Journal Experts (AJE) (https://www.aje.com/ ) is one such service that has extensive experience helping authors meet PLOS guidelines and can provide language editing, translation, manuscript formatting, and figure formatting to ensure your manuscript meets our submission guidelines. Please note that having the manuscript copyedited by AJE or any other editing services does not guarantee selection for peer review or acceptance for publication. 

A clean copy of the edited manuscript (uploaded as the new *manuscript* file)”.

3. In the online submission form, you indicated that [The data underlying the results presented in the study are available from the corresponding author and possible to reach through the following email: getish98@gmail.com]. 

5. We notice that your supplementary figures are uploaded with the file type 'Figure'. Please amend the file type to 'Supporting Information'. Please ensure that each Supporting Information file has a legend listed in the manuscript after the references list.

Reviewers' comments:

Reviewer's Responses to Questions

**Comments to the Author**

1. Is the manuscript technically sound, and do the data support the conclusions?

Reviewer #1: Yes

Reviewer #2: Partly

2. Has the statistical analysis been performed appropriately and rigorously?

Reviewer #1: Yes

Reviewer #2: I Don't Know

3. Have the authors made all data underlying the findings in their manuscript fully available?

Reviewer #1: Yes

Reviewer #2: Yes

4. Is the manuscript presented in an intelligible fashion and written in standard English?

Reviewer #1: Yes

Reviewer #2: No

5. Review Comments to the Author

Reviewer #1: Comments to the Author

In this study, Belete et al. did the study focusing on Prevalence and risk factors of diabetes among adult populations of Hawassa town, Ethiopia: A community based cross-sectional study.

The main goal of the study was to fill the gaps in existing research on the prevalence of diabetes and its predictors. The findings of this research could improve the methods used to prevent and manage diabetes. Furthermore, the lack of data from African countries highlights the importance of the data presented by Belete et al., as it contributes valuable information to the current literature. As a result, I have several suggestions that the authors may consider to improve the quality of the manuscript.

Title page

Please! Remove the prefix: prof, from HL Sithole

It is my belief that the primary author possesses two addresses, therefore it is imperative that both are included on the title page in the following manner:

Getu Kassa Belete1, 2*, HL Sithole2

1World Health organization (WHO), Sidama, Ethiopia

2College of Health Studies at the University of South Africa (UNISA)

*Corresponding Author

E-mail: getish98@gmail.com

Abstract part

CI, AOR, DM: Please! Describe fully prior to abbreviating the words

Being illiterate: it is better, if you say “unable to read and write” than saying “illiterate”

Keywords: please! Remove the second keywords “Key words: Magnitude, diabetes, non-communicable, risk factors”

Please! Use similar writing manner when writing: medium cycle (AOR: 2.79; 95% CI: 1.02, 7.63), having ever chewed khat (AOR: 6.50; 95% CI: 4.07, 10.39)

Conclusion part: authors should focus on modifiable risk factors: such as lifestyle changing (dietary habit, declining options of overweight and obesity) and behavioural changes. Because risk factors in your study were almost modifiable risks except gender.

Introduction Part

The authors would benefit from including the diabetes trends observed in previous studies conducted in the Sidama region.

Q1) Before mentioning the objective of their study, it is crucial to emphasize the significant gaps that the authors identified in previous research, which motivated them to address these gaps in their own study.

Materials and methods

Study setting part

The study was conducted in Hawassa town, Ethiopia. Modify it to: “The study was conducted in Hawassa town, Sidama regional state, Southern Ethiopia”

Inclusion and exclusion part

Kindly provide clarification on the cases of previously diagnosed and known diabetes, as there is no information regarding them in the inclusion/exclusion section.

Sample size determination part

Q2) Why did the authors choose to rely solely on the prevalence rate when determining the sample size, given that their objective was to determine both prevalence and risk factors? Why didn't the authors attempt to determine the sample size using risk factors? It is advisable to calculate the sample size using both the prevalence rate and risk factors, and then select the larger sample size as the final sample size by comparing the sample sizes calculated using both methods.

Study variables part

Q3)Why the economic/monthly income status of the study subjects was not considered as an independent factor?

Q4) Why the study did not take into account independent factors like comorbid health conditions, hypertension, or the use of medications for comorbid health problems among the study subjects?

Definitions part

Body Mass Index (BMI): A standard cut-offs limit calculated using the formula Kg/m2 to assess

an individual’s fat.

Please! Correct it to: “Body Mass Index (BMI): A standard cut-offs limit calculated using the formula Kg/m2 to assess an individual’s fat

Data collection part

A standard glucometer (Auto-cad) used for fasting blood test following overnight fasting prior to testing.

Q5) is it auto code or auto-cad?

“The data collection period was from September, 2023 to November, 2023”.

Please! Remove this redundant sentence because once it was mentioned in the study period part.

Q6) How was the accuracy of the glucometer instrument checked at first and at regular intervals during data collection?

Q7) The management of data quality remains unclear as there is no information provided on how it was addressed?

Q8) What actions did you or your study team take for the newly diagnosed DM subjects? It is important to address this aspect as there is no information provided about these subjects in your data collection or ethical issue sections

Results part

Prevalence of diabetes

Q9) Out of the total prevalence rate of 14.4%, what is the number of newly diagnosed cases of DM and how many are cases of previously diagnosed and known DM?

Q10) what is the prevalence rate of prediabetes in your study?

*It is preferable to provide an explanation regarding the “Behavioural and biological risk factors of diabetes and tis table” prior to discussing the prevalence of diabetes.

*please! Put footnote for all abbreviated words below table (table3: COR, AOR, CI)

Discussion part

*Slightly higher than a study done in Hawassa zuria woredas (12.2%) [17]. Not slightly higher but comparable because your study confidence interval included the rate. The rreference 17 reveals the prevalence of DM in Hawasaa zuria is not 12.2% but 12.4%. Please check and correct it.

*Also, the highest prevalence of diabetes (19.7%) and (18.4%) were reported in the age category between 50-59 years and 30-39 years, respectively. Please! Use the word higher but not highest

Q11) what is possible reason for this increment? Please! Mention it.

*In this particular research, the prevalence of diabetes was higher in Males than females,

20.7% versus 10.0% respectively. The finding is higher than the study done in Korea (males,

10.7% and females, 8.4%) [19] and Ethiopia (males, 9.1% and females, 8.4%) [20].

What are your possible reasons for the variations? Please describe it. Please! Use similar discussion manner in all discussion parts.

Reviewer #2: I thank the editor for his/her invitation and the authors for their efforts to come up with this interesting scientific report. I suggest and inquire the authors to make their report more robust and readable.

“…risk factors…” in the title should be replaced by ‘associated factor’.

The authors included the population between ages 20 to 69? What were your rationale behind the inclusion this particular population.

The authors stressed description of type-2 diabetes in the introduction section. Was their study only assessed the prevalence of type-2 diabetes?

The introduction section did not show your depth of knowledge about the risk factors of diabetes, your second specific objective.

The study setting is not detail. The number of sub-cities and the estimated number of house-holds would be included.

The authors’ excluded pregnant women, physical disabilities, terminally sick and individual with mental illness. Why did you exclude all physical disabled and mentally ill individuals? It seems like discrimination. You don’t think people with physical disability and mental problems develop diabetes in their lives? Also, known diabetic women could get pregnant and still might be excluded from this study.

How did you manage when you got more than one eligible individuals (aged 20 to 69) in the selected house-holds?

Under “variables” subheading, it would better to say ‘prevalence of diabetes’

Term definitions would be cited.

“Currently tobacco smoking” this would be time bounded. What if an individual had been smoked until the last month prior to your survey?

The authors measured the BMI of the participants but they did not detail the measurement instrument and procedures as well as the instrument’s calibres. How did you manage waste disposal throughout the study period?

Authors did not mention the data quality control measures they were applied. Did you check the tool’s validity? How did you ensure whether the subjects fasted over night? At what time in the morning you tested their blood sugar? If you executed it before the subject’s breakfast, you could not test the blood sugar of that all participants within two months. This issue needs explanation.

The authors stated that they obtained a written informed consent from the study subjects. So how did they obtained the consent from 173 uneducated (they said “illiterate” that is also uncomfortable name) participants?

It would be better if table 3 was not congested. To do so, the p-values of the bivariable analysis could be omitted or included in the text narration. The NA variables in this table also would not be included. If they did not show association in the bivariable analysis why did you included them in this table?

The authors did not discuss all their findings.

The authors would state the shortcomings of their study.

6. PLOS authors have the option to publish the peer review history of their article (what does this mean? ). If published, this will include your full peer review and any attached files.

**Do you want your identity to be public for this peer review?** For information about this choice, including consent withdrawal, please see our Privacy Policy .

Reviewer #1: **Yes: ** Agete Tadewos Hirigo

Reviewer #2: No

---

## [Author Response · Author response to Decision Letter 0]

10 Sep 2024

Dear Editor

We appreciate you looking over our manuscript and providing us with this valuable input. We made changes to the manuscript in response to the comments, and one of the attached manuscript files shows the highlighted changes. Point by point, we offer our answer, the questions, and our response as follows:

Comments on Journal requirements:

Title page

Q1. Please! Remove the prefix: prof, from HL Sithole

It is my belief that the primary author possesses two addresses, therefore it is imperative that both are included on the title page in the following manner:

Getu Kassa Belete1, 2*, HL Sithole2

1World Health organization (WHO), Sidama, Ethiopia

2College of Health Studies at the University of South Africa (UNISA)

*Corresponding Author

E-mail: getish98@gmail.com

Answer for Q1. Thank you for this important comments. We have removed the prefix and also indicated the two addresses per comment on page 1.

Abstract part

Q2. CI, AOR, DM: Please! Describe fully prior to abbreviating the words

Answer for Q2. We appreciate your feedback. Every acronym was fully explained per the comments on page 2 & 3.

Q3. Being illiterate: it is better, if you say “unable to read and write” than saying “illiterate”

Answer for Q3. Thank you for the comment. As per the comment changed illiterate to unable to read and write on page 2,14,18 and 19.

Q4. Keywords: please! Remove the second keywords “Key words: Magnitude, diabetes, non-communicable, risk factors”

Answer for Q4. Thank you for the feedback. We removed the second keywords per the comment on page 3.

Q5. Please! Use similar writing manner when writing: medium cycle (AOR: 2.79; 95% CI: 1.02, 7.63), having ever chewed khat (AOR: 6.50; 95% CI: 4.07, 10.39)

Answer for Q5. Thank you for the comment. We corrected per your feedback on page 18.

Q6. Conclusion part: authors should focus on modifiable risk factors: such as lifestyle changing (dietary habit, declining options of overweight and obesity) and behavioural changes. Because risk factors in your study were almost modifiable risks except gender.

Answer for Q6. We appreciate for your invaluable comment. We included the feedback in the conclusion part on page 3 and 22.

Introduction Part

The authors would benefit from including the diabetes trends observed in previous studies conducted in the Sidama region.

Q1. Before mentioning the objective of their study, it is crucial to emphasize the significant gaps that the authors identified in previous research, which motivated them to address these gaps in their own study.

Answer to Q1. We appreciate your feedback. In response to your comment on page 5, We have incorporated earlier national and regional research findings.

Materials and methods

Study setting part

The study was conducted in Hawassa town, Ethiopia. Modify it to: “The study was conducted in Hawassa town, Sidama regional state, Southern Ethiopia”

Answer. We appreciate for the comment. We included "Sidama regional state, Southern Ethiopia" per the comment on page 6.

Answer

Inclusion and exclusion part

Kindly provide clarification on the cases of previously diagnosed and known diabetes, as there is no information regarding them in the inclusion/exclusion

Answer. Thank you for the comment. Previously diagnosed and known diabetic patients were included in the study. Please see in the inclusion and exclusion part on page 6.

Sample size determination part

Q2. Why did the authors choose to rely solely on the prevalence rate when determining the sample size, given that their objective was to determine both prevalence and risk factors? Why didn't the authors attempt to determine the sample size using risk factors? It is advisable to calculate the sample size using both the prevalence rate and risk factors, and then select the larger sample size as the final sample size by comparing the sample sizes calculated using both methods.

Study variables part.

Answer Q2. We appreciate for the feedback to consider and compare sample size determination for various associated factors. Accordingly, it was calculated as shown on table , on page 8.

Q3. Why the economic/monthly income status of the study subjects was not considered as an independent factor?

Answer for Q3. We appreciate your feedback. Since this research is self-sponsored, it would have required a significant investment of resources to incorporate all variables, so we only took a few of the WHO STEP surveillance guideline variables into consideration.

Q4. Why the study did not take into account independent factors like comorbid health conditions, hypertension, or the use of medications for comorbid health problems among the study subjects?

Answer for Q4. We appreciate your comment. We included the analysis findings of hypertension on page 17.

Definitions part

Body Mass Index (BMI): A standard cut-offs limit calculated using the formula Kg/m2 to assess an individual’s fat. Please! Correct it to: “Body Mass Index (BMI): A standard cut-offs limit calculated using the formula Kg/m2 to assess an individual’s fat

Answer: We appreciate your comment. We corrected and incorporated Kg/m2 in the manuscript where necessary.

Data collection part

A standard glucometer (Auto-cad) used for fasting blood test following overnight fasting prior to testing.

Q5. is it auto code or auto-cad?

Answer for Q5. Thank you for the comment. It is auto-code. We corrected and incorporated in the manuscript accordingly.

“The data collection period was from September, 2023 to November, 2023”.

Please! Remove this redundant sentence because once it was mentioned in the study period part.

Answer: We appreciate for the feedback. I corrected redundant sentences of the study period in the manuscript where necessary.

Q6. How was the accuracy of the glucometer instrument checked at first and at regular intervals during data collection?

Answer for Q6. We appreciate your feedback. Calibration was done using glucometer code key both before the blood test and on a frequent basis during the data collection period.

Q7. The management of data quality remains unclear as there is no information provided on how it was addressed?

Answer for Q7. Data was submitted to supervisors and checked for completeness and accuracy on daily base. Moreover, prior to analysis data was checked, cleaned and entered in SPSS to ensure data quality.

Q8) What actions did you or your study team take for the newly diagnosed DM subjects? It is important to address this aspect as there is no information provided about these subjects in your data collection or ethical issue sections

Answer for Q8. Thank you for the comment. The newly diagnosed diabetes patients were advised on lifestyle modification. Moreover, they were told to make medical follow up at the nearby health facility at regular base.

Results part

Prevalence of diabetes

Q9. Out of the total prevalence rate of 14.4%, what is the number of newly diagnosed cases of DM and how many are cases of previously diagnosed and known DM?

Answer for Q9. The number of newly diagnosed DM patients were 101 (63.1%). The number of known DM cases were 59 (36.9%)

Q10. what is the prevalence rate of prediabetes in your study?

*It is preferable to provide an explanation regarding the “Behavioural and biological risk factors of diabetes and its table” prior to discussing the prevalence of diabetes.

*please! Put footnote for all abbreviated words below table (table3: COR, AOR, CI)

Answer for Q10. Thank you for the comments. The prevalence of pre-diabetes was 283 (26.9%) among study participants excluding known diabetes patients from the denominator. Behavioural and biological risk factors of diabetes were discussed prior to prevalence of diabetes per your comment. Likewise, Abbreviations were explained on page 20 below table 4.

Discussion part

*Slightly higher than a study done in Hawassa zuria woredas (12.2%) [17]. Not slightly higher but comparable because your study confidence interval included the rate. The reference 17 reveals the prevalence of DM in Hawassa zuria is not 12.2% but 12.4%. Please check and correct it.

Answer: Thank you for the comment. We corrected and changed to 12.4%, please see on page 20.

*Also, the highest prevalence of diabetes (19.7%) and (18.4%) were reported in the age category between 50-59 years and 30-39 years, respectively. Please! Use the word higher but not highest

Answer: We appreciate for the comment. We changed "highest" to "higher" per your comment. Please see on page 20.

Q11. what is possible reason for this increment? Please! Mention it.

*In this particular research, the prevalence of diabetes was higher in Males than females,

20.7% versus 10.0% respectively. The finding is higher than the study done in Korea (males,

10.7% and females, 8.4%) [19] and Ethiopia (males, 9.1% and females, 8.4%) [20].

What are your possible reasons for the variations? Please describe it. Please! Use similar discussion manner in all discussion parts.

Answer for Q11: Thank you for the feedback. We included possible reasons please see on page 21.

Reviewer #2: I thank the editor for his/her invitation and the authors for their efforts to come up with this interesting scientific report. I suggest and inquire the authors to make their report more robust and readable.

“…risk factors…” in the title should be replaced by ‘associated factor’.

Answer: We appreciate for the comment. We replaced risk factor by associated factor in the title page as per the comment please see on page 1.

The authors included the population between ages 20 to 69? What were your rationale behind the inclusion this particular population?

Answer: The population 20 to 69 years were included in this study for proportional sample size allocation for the age category based on the national statistical report.

The authors stressed description of type-2 diabetes in the introduction section. Was their study only assessed the prevalence of type-2 diabetes?

Answer: Thank you for the comment. The study focused primarily prevalence of diabetes and associated factors among adult population in Hawassa town which could be both type II and type I diabetes. Per the comment, prevalence of type I diabetes was incorporated in the introduction part of the manuscript please see on page 3 and 4.

The introduction section did not show your depth of knowledge about the risk factors of diabetes, your second specific objective.

Answer: Thank you for your comment. We have incorporated risk factors associated with diabetes in the introduction section in detail per your comment please see on page 4 and 5.

The study setting is not detail. The number of sub-cities and the estimated number of house-holds would be included.

Answer. Thank you for the feedback. We included the number of sub-cities, villages (kebeles) and estimated households in the town please see on page 6 .

The authors’ excluded pregnant women, physical disabilities, terminally sick and individual with mental illness. Why did you exclude all physical disabled and mentally ill individuals? It seems like discrimination. You don’t think people with physical disability and mental problems develop diabetes in their lives? Also, known diabetic women could get pregnant and still might be excluded from this study.

Answer. We appreciate for the comments. It is apparent that all individuals could be at risk of having diabetes if exposed to risk factors. However, pregnant women were excluded in this study because they might have gestational diabetes which may result in prevalence bias. Furthermore, while taking weight still would reflect bias on body mass index findings. Likewise, Patients with mental illness were excluded because they were not eligible to provide correct information during interview. Also, physical disabilities need further care with special instrument support while taking multiple measurements. Due to these facts, they were not included in this study.

How did you manage when you got more than one eligible individuals (aged 20 to 69) in the selected house-holds?

Answer. We used kish method (random selection of study subjects from eligible age groups at household level) to select a single individual when there were more than one eligible individuals in one household.

Under “variables” subheading, it would better to say ‘prevalence of diabetes’

Term definitions would be cited.

Answer. Thank you for the comment. We included and corrected as prevalence of diabetes per the comment please see on page 8.

“Currently tobacco smoking” this would be time bounded. What if an individual had been smoked until the last month prior to your survey?

Answer. Thank you for the comment. All level of smoking status including past smoking history were included in the study as shown on Table 3, please see on page 16. However, in this particular study “current smoking" fulfils criteria for multi variable analysis.

The authors measured the BMI of the participants but they did not detail the measurement instrument and procedures as well as the instrument’s calibers. How did you manage waste disposal throughout the study period?

Answer: Thank you for the comments. Measurements were taken using a portable digital weight scale and height measuring instrument (Seca). Daily calibration was carried out with a standard kilogram weight. In addition, the instrument was positioned nearer a wall on a level floor. The device was tested for satisfactory operation, pointing at zero reading, before any data was collected. In addition, study participants were instructed to remove overcoats and jackets, stand erect on a beam facing the data collectors, and go barefoot. The correct protocol for disposing of waste was adhered to when gathering the data. Waste materials were collected in plastic bags and disposed of at the neighboring health facilities' incinerators for burning.

Authors did not mention the data quality control measures they were applied. Did you check the tool’s validity? How did you ensure whether the subjects fasted over night? At what time in the morning you tested their blood sugar? If you executed it before the subject’s breakfast, you could not test the blood sugar of that all participants within two months. This issue needs explanation.

Answer: I appreciate your feedback. Experts with extensive experience evaluated the tool's validity, and questions that were deemed unclear were adjusted accordingly. Additionally, on a pre-test, 5% of the sample size was examined. Because the questioner measured what it was supposed to measure, it was valid. We tested before the study participants eat their breakfast. Additionally, data collectors strongly recommended fasting for the entire night. Additionally, it was often questioned if individuals had fasted before the exam. We recommended that research participants take a HgbA1c test two to three months later. However, due to resource constraints, we do not conduct testing after two to three months. Please see in the data collection section on page 9 and 10.

The authors stated that they obtained a written informed consent from the study subjects. So how did they obtained the consent from 173 uneducated (they said “illiterate” that is also uncomfortable name) participants?

Answer. Thank you for the comment. Study subjects who were unable to read and write gave their consent with their finger print after the consent form was read and explained. Please, see in ethical consideration section on page 12.

It would be better if table 3 was not congested. To do so, the p-values of the bivariable analysis could be omitted or included in the text narration. The NA variables in this table also would not be included. If they did not show association in the bivariable analysis why did you included them in this table?

Answer: Thank you for the feedback. We removed all the p values for both bi variable and multivariable analysis. Also, variables with not candidate for multi variable analysis were removed per your invaluable comment. See table 4. Kindly be informed that name table 3 is changed to table 4.

The authors did not discuss all their findings.

Answer: Thank you for your feedback. All the prevalence of diabetes mellitus and all variables having association wit

---

## [Decision Letter · Decision Letter 1]

23 Dec 2024

PONE-D-24-07197R1Prevalence and risk factors of diabetes among adult populations of Hawassa town, Ethiopia: A community based cross-sectional studyPLOS ONE

Dear Dr. Belete,

Thank you for submitting your manuscript to PLOS ONE. After careful consideration, we feel that it has merit but does not fully meet PLOS ONE’s publication criteria as it currently stands. Therefore, we invite you to submit a revised version of the manuscript that addresses the points raised during the review process.

We look forward to receiving your revised manuscript.

Kind regards,

Mengistu Hailemariam Zenebe, PhD

Academic Editor

PLOS ONE

Journal Requirements:

Reviewers' comments:

Reviewer's Responses to Questions

**Comments to the Author**

1. If the authors have adequately addressed your comments raised in a previous round of review and you feel that this manuscript is now acceptable for publication, you may indicate that here to bypass the “Comments to the Author” section, enter your conflict of interest statement in the “Confidential to Editor” section, and submit your "Accept" recommendation.

Reviewer #1: All comments have been addressed

2. Is the manuscript technically sound, and do the data support the conclusions?

Reviewer #1: Yes

3. Has the statistical analysis been performed appropriately and rigorously?

Reviewer #1: Yes

4. Have the authors made all data underlying the findings in their manuscript fully available?

Reviewer #1: Yes

5. Is the manuscript presented in an intelligible fashion and written in standard English?

Reviewer #1: Yes

6. Review Comments to the Author

Reviewer #1: Manuscript Number: PONE-D-24-07197R1 "Prevalence and risk factors of diabetes among adult populations of Hawassa town, Ethiopia: A community based cross-sectional study"

Dear authors,

Thank you for your prompt response and for making significant revisions to your manuscript. I appreciate the effort you have put into improving the work so far. After reviewing the latest version, I have added a few additional comments and suggestions that I believe will further enhance the quality of the manuscript. These revisions are intended to help clarify certain aspects and make your findings more insightful for the readers. Once these changes are incorporated, I believe the manuscript will be well-suited for publication.

C1.Title: Please! Correct it to “Prevalence and associated factors of diabetes among adult populations of Hawassa town, southern Ethiopia. A community based cross-sectional study”

Abstract part

C2. Background

Therefore, the aim of this study was to investigate the magnitude and preventable risk factors of diabetes among adult populations in Hawassa town, Ethiopia

But in your included non-modifiable risks like sex please modify it to “Therefore, the aim of this study was to investigate the magnitude and risk factors of diabetes among adult populations in Hawassa town, southern Ethiopia” “Preventable risk factors” Please! Correct it to “modifiable risk factors” because modifiable risk factors" is generally the better term to use when discussing factors that can be changed or controlled to reduce the risk of disease or adverse outcomes.

C3.A community based cross-sectional study was conducted from September, 2023 to November, 2023 among adult populations” avoid redundant words “Hawassa town, Ethiopia” because the words described in the abstract introduction section.

C4. Methods If you are interested, you can use this "An interviewer-administered questionnaire was used to collect data. Additionally, participants were also instructed to fast overnight, after which the standard fasting blood sugar test was conducted. A binary logistic regression model was fitted to identify independent predictors of diabetes."

C5. Conclusion If you are comfortable, you can use this “This study identified a high prevalence of diabetes among adults in Hawassa, driven by various risk factors. This presents an opportunity to mitigate diabetes risk through public health measures, including avoiding khat chewing, promoting healthy diets, managing overweight and obesity, implementing community-based screening, enhancing health literacy, and integrating health information into daily life.”

Introduction part

C6: Also, the magnitude of khat chewing in Hossana, Ethiopia reported at 58%, of those khat chewers, 75.2% and 24.7% were men and women, respectively [19]. It would be more informative if the authors presented data on the prevalence of diabetes/prediabetes and examined the association between khat chewing and the increased prevalence of these conditions. The above statement only addresses the magnitude of khat chewing without linking it to blood sugar levels or diabetes risk. You can use the following 2 or 3 literatures from the following 3 links ".

1. https://pmc.ncbi.nlm.nih.gov/articles/PMC7024885/pdf/dmso-13-307.pdf

2. file:///C:/Users/admin/Downloads/Prevalence_and_associated_risk_factors_of_pre-diab.pdf

3. https://www.dovepress.com/article/download/71642

C7. Introduction part last sentence: please! Modify it to “A previous study conducted in Hawassa town was facility-based and mainly focused on individuals seeking medical care. In contrast, this study was aimed to determine the prevalence of diabetes and risk factors among adults through a community-based survey in Hawassa town, southern Ethiopia”

Materials and methods part

Study setting

C8. There are also 81,523 households in the town

Study design and period

C9. You mentioned only the study design but not period and please! Correct it to “A community based cross-sectional study design was employed from September, 2023 to November, 2023 in Hawassa town, southern Ethiopia.”

Definition of terms part

C10. Pls! Correct it to “Body Mass Index (BMI): A standard cut-offs limit calculated using the formula Kg/m2 to assess an individual’s body mass and classified as underweight (<18.5kg/m²), normal weight (18.5–24.9kg/m²), overweight (25–29.9kg/m²), and obesity (≥30kg/m²). Reference: A healthy lifestyle - WHO recommendations. https://www.who.int/europe/news-room/fact-sheets/item/a-healthy-lifestyle---who-recommendations

C11. Diabetes: Pls! correct it to “It is defined as a fasting blood glucose level of ≥126 mg/dL (or ≥7.0 mmol/L), a self-reported diagnosis of diabetes, or the use of oral or injectable hypoglycemic agents."

C12. Fasting blood test: please! Correct it to “Fasting blood glucose test”

C13. Please! Minimize Ethical consideration as much as possible.

Discusion part

("Please refine the Discussion section to make it more insightful.") Use the below paragraph as a sample

C14. "The finding is consistent with previous community-based studies conducted in southern Ethiopia, such as the 14.7% prevalence in the ostracized Menja community [25] and 12.4% in the Hawassa Zuria Woreda [29]." In contrast, the finding is higher than those of previous studies conducted in other countries, such as Botswana (9.3%), Zanzibar (4.4%), and Thailand (9.9%) [21, 22, 31]."

Conclusion part:

If you are interested use this “In conclusion, this study identified a noticeable prevalence of diabetes and its association with modifiable risk factors among the adult population of Hawassa town. The evidence-based risk factors associated with diabetes highlight the need for urgent public health interventions, particularly community-level screening programs aimed at early detection and prevention.in addition, the findings emphasize the importance of addressing behavioral practices, such as khat chewing, and promoting lifestyle modifications, including dietary changes and the reduction of overweight and obesity. Regular health check-ups and follow-up care for individuals with diabetes are also crucial. Moreover, risk communication strategies for diabetes should be integrated into public health campaigns to raise awareness. Lastly, these findings could serve as a baseline for future studies focused on operational study related to diabetes management and prevention.”

7. PLOS authors have the option to publish the peer review history of their article (what does this mean? ). If published, this will include your full peer review and any attached files.

**Do you want your identity to be public for this peer review?** For information about this choice, including consent withdrawal, please see our Privacy Policy .

Reviewer #1: **Yes: ** Agete Tadewos Hirigo

---

## [Author Response · Author response to Decision Letter 1]

7 Jan 2025

Dear Editor,

We have responded to all comments given by the editor and reviewers . We are also ready to respond to any comments if further required as deemed necessary.

---

## [Editor Report · Decision Letter 2]

10 Jan 2025

Prevalence and associated  factors of diabetes among adult populations of Hawassa town, southern Ethiopia: A community based cross-sectional study

PONE-D-24-07197R2

Dear Dr. Belete,

We’re pleased to inform you that your manuscript has been judged scientifically suitable for publication and will be formally accepted for publication once it meets all outstanding technical requirements.

Kind regards,

Mengistu Hailemariam Zenebe, PhD

Academic Editor

PLOS ONE
---

## [Editor Report · Acceptance letter]

PONE-D-24-07197R2

PLOS ONE

Dear Dr. Belete,

I'm pleased to inform you that your manuscript has been deemed suitable for publication in PLOS ONE. Congratulations! Your manuscript is now being handed over to our production team.

Kind regards,

on behalf of

Dr. Mengistu Hailemariam Zenebe

Academic Editor

PLOS ONE